# ADAPTIVE UPDATE DIRECTION RECTIFICATION FOR UNSUPERVISED CONTINUAL LEARNING

## ABSTRACT

Recent works on continual learning have shown that unsupervised continual learning (UCL) methods rival or even beat supervised continual learning methods. However, most UCL methods typically adopt fixed learning strategies with pre-defined objectives and ignore the influence of the constant shift of data distributions on the newer training process. This non-adaptive paradigm tends to achieve sub-optimal performance, since the optimal update direction (to ensure the trade-off between old and new tasks) keeps changing during training over sequential tasks. In this work, we thus propose a novel UCL framework termed AUDR to adaptively rectify the update direction by a policy network (i.e., the Actor) at each training step based on the reward predicted by a value network (i.e., the Critic). Concretely, different from existing Actor-Critic based reinforcement learning works, there are three vital designs that make our AUDR applicable to the UCL setting: (1) A reward function to measure the score/value of the currently selected action, which provides the ground-truth reward to guide the Critic's predictions; (2) An action space for the Actor to select actions (i.e., update directions) according to the reward predicted by the Critic; (3) A multinomial sampling strategy with a lower-bound on the sampling probability of each action, which is designed to improve the variance of the Actor's selected actions for more diversified exploration. Extensive experiments show that our AUDR achieves state-of-the-art results under both the in-dataset and cross-dataset UCL settings. Importantly, our AUDR also shows superior performance when combined with other UCL methods, which suggests that our AUDR is highly extensible and versatile.

## 1    INTRODUCTION

Continual learning has recently drawn great attention, for it can be applied to learning on a sequence of tasks without full access to the historical data (Rusu et al., 2016; Rebuffi et al., 2017; Lopez-Paz & Ranzato, 2017; Fernando et al., 2017; Kirkpatrick et al., 2017; Zenke et al., 2017). Most of existing methods focus on supervised continual learning (SCL), and only a few (Rao et al., 2019; Madaan et al., 2021; Fini et al., 2022) pay attention to unsupervised continual learning (UCL). UCL is an important yet more challenging task which requires a model to avoid forgetting previous knowledge after being trained on a sequence of tasks without labeled data.

Recent UCL methods (Rao et al., 2019; Madaan et al., 2021; Fini et al., 2022) have achieved promising results, and even outperform the SCL methods. However, these UCL methods are still limited by the fixed learning strategies with pre-defined objectives. For instance, LUMP (Madaan et al., 2021) proposed a fixed lifelong mixup strategy that integrates current and memory data in a random ratio sampled from a Beta distribution regardless of the shift in data distributions. This non-adaptive paradigm is not ideal for UCL, since the optimal update direction of achieving the best performance on all learned tasks is continuously changing during training. Therefore, a new adaptive paradigm for UCL to model the process of selecting the optimal update direction is needed.

In this work, we thus devise a new UCL framework termed AUDR that can adaptively rectify the update direction (see Figure 1), where a policy network (i.e., the Actor) is proposed to select the best action for current data batch and a value network (i.e., the Critic) is designed for predicting the action's latent value. The Actor is trained to maximize the Critic's prediction reward, and the Critic is trained to more precisely predict the reward for the Actor's selected action. Different from the

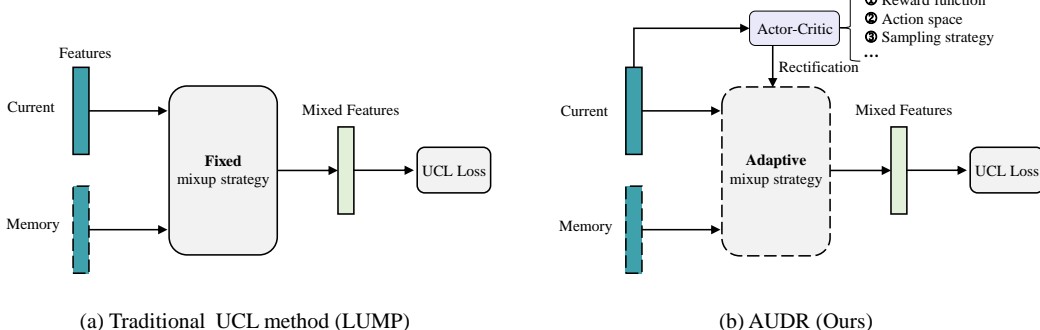

(a) Traditional  UCL method (LUMP)  (b) AUDR (Ours)

Figure 1: Illustration of the traditional UCL method LUMP and our AUDR. Their main difference lies in that our AUDR adopts an Actor-Critic architecture to rectify the update direction (i.e., adaptive mixup strategy) with three core designs while LUMP has a fixed mixup strategy.

short-sighted approaches (e.g., directly using a learnable parameter) that can only adjust the update direction based on current batch of data/loss, our AUDR predicts the total future rewards which is more and more precise/reliable during training. This is inspired by the Actor-Critic learning, a combination of policy-based and value-based methods, in the reinforcement learning field (Sutton et al., 1999; Haarnoja et al., 2018; Yarats et al., 2020; Mu et al., 2022). Actor-Critic learning enables the policy network to be updated at each training step (instead of after completing each task) with sampled transitions (i.e., from one state to next state), and thus it is possible to be transferred to UCL. However, we still have difficulty in directly deploying existing Actor-Critic methods under the UCL setting, because: (1) there is no environment (or reward function) that could give feedback rewards to all input states; (2) the action space for the Actor is not explicitly defined; (3) how to alleviate the problem of the trade-off between the old and new tasks remains unclear.

To address these problems, our AUDR thus has three core designs: (1) A reward function defining the ground-truth reward to guide the Critic's training. It is based on two UCL losses of the next model (after one-step gradient descent) by respectively using the current and memory data. Thus, this reward can represent the changes of model performance on old and new tasks with the selected action, which is then utilized to conduct a continual TD-error loss to train the Critic. (2) An action space containing different actions (i.e., different update directions) for the Actor to select. To be more specific, an action with larger value (e.g., 0.99) represents that the memory data accounts for higher percentage when mixed with current data, and thus the update direction is more oriented towards improving the model performance on old tasks. (3) A multinomial sampling strategy to sample the action based on the action probability distributions predicted by the Actor. Concretely, for each input feature, the Actor outputs a probability distribution where each action is associated with a probability holding a lower-bound above zero to improve the variance of the Actor's selected actions. We then multinomially sample one action per feature and all samples vote for the final action. This strategy is designed to explore more diverse actions to avoid the model falling into a local optimal update direction. Note that the Actor-Critic module of our AUDR is only employed for training and we only use the backbone network for testing as in LUMP (Madaan et al., 2021).

Furthermore, we combine the proposed adaptive paradigm with another representative method DER (Buzzega et al., 2020) for UCL to verify the extensibility of our AUDR. Specifically, we use different coefficients of the penalty loss as the new action space to replace the action space mentioned above, which is a key factor that affects the update direction in DER. Other settings remain the same as in our original AUDR. We find that our AUDR+DER outperforms DER for UCL by a large margin. This demonstrates that our AUDR is highly generalizable/versatile. We believe that our work could bring some inspirations to the continual learning community.

Our main contributions are four-fold: (1) We are the first to deploy an adaptive learning paradigm for UCL, i.e., we propose a novel UCL framework AUDR with an Actor-Critic module. (2) We devise three core designs in our AUDR to ensure that the Actor-Critic architecture is seamlessly transferred to UCL, including a reward function, an action space, and a multinomial sampling strategy. (3) Extensive experiments on three benchmarks demonstrate that our AUDR achieves new state-of-the-art results on UCL. (4) Further analysis on combining our proposed adaptive paradigm with another UCL method shows that our AUDR is highly generalizable and has great potential in UCL.

## 2 RELATED WORK

**Continual Learning.** Existing continual learning methods can be mainly categorized into three groups: (1) *Expansion-based* methods have dynamic architectures which add extra extended networks for in-coming new tasks (Rusu et al., 2016; Fernando et al., 2017; Alet et al., 2018; Chang et al., 2018; Li et al., 2019). (2) *Regularization-based* methods deploy either regularization constraints (Li & Hoiem, 2017; Aljundi et al., 2017; Hou et al., 2018; Rannen et al., 2017; Hou et al., 2019) or penalty losses (Kirkpatrick et al., 2017; Zenke et al., 2017; Farajtabar et al., 2020; Saha et al., 2021) to align the old and new models. (3) *Rehearsal-based* methods adopt a memory buffer to restore the memory data of previous tasks (Rebuffi et al., 2017; Lopez-Paz & Ranzato, 2017; Riemer et al., 2018; Chaudhry et al., 2019; Buzzega et al., 2020; Aljundi et al., 2019; Chaudhry et al., 2020; Cha et al., 2021). In contrast to these supervised continual learning works, (Rao et al., 2019; Madaan et al., 2021; Fini et al., 2022; Davari et al., 2022) have started to study the unsupervised continual learning (UCL) setting in a fixed learning paradigm by combining continual learning methods with unsupervised methods (Chen et al., 2020; Chen & He, 2021; Zbontar et al., 2021). Our AUDR is basically a rehearsal-based method for UCL but with an adaptive training paradigm which rectifies the update direction at each training step. Note that the adaptive paradigm of our AUDR is different from those methods that utilize a learnable parameter to adjust the update direction or even learn to prompt (Wang et al., 2022), since our AUDR is long-sighted (predicting future rewards) and the adaptively rectified action is only used for training (the Actor-Critic module is dropped at the test phase). We provide more discussions about their differences in Appendix B.

**Actor-Critic Learning.** Actor-Critic is a widely-used architecture in recent reinforcement learning (RL) works. (Peters & Schaal, 2008) builds the Actor-Critic algorithm on standard policy gradient formulation to update the Actor; (Schulman et al., 2017; Mnih et al., 2016; Gruslys et al., 2017; Haarnoja et al., 2018) choose to maximize or regularize the entropy of the policy; CURL (Laskin et al., 2020) combines the unsupervised learning with Actor-Critic reinforcement learning; DrQ (Yarats et al., 2020) designs a data-regularized Q to improve the Actor-Critic method; CtrlFormer (Mu et al., 2022) proposes a control transformer to tackle the forgetting problem in visual control. Our AUDR is the first work to apply Actor-Critic learning to the UCL setting. It consists of a similar Actor-Critic module as in CURL, DrQ, and CtrlFormer, but also has vital differences: a reward function for UCL to replace the environment of RL, a new action space designed for the Actor, and a multinomial sampling strategy to ensure the diversity of actions.

## 3 METHODOLOGY

### 3.1 UNSUPERVISED CONTINUAL LEARNING

Unsupervised continual learning (UCL) requires the model to be trained on a sequence of tasks without labeled data. We follow the learning protocol proposed in LUMP (Madaan et al., 2021) to conduct our study on UCL. Concretely, let $\mathcal{D} = [\mathcal{D}_1, \mathcal{D}_2, \cdots, \mathcal{D}_N]$ denotes a dataset with $N$ tasks. For each task $t$, it has $\mathcal{D}_t = \{x_{t,i}, y_{t,i}\}_{i=1}^{n_t}$ with $n_t$ samples, where $x_{t,i}$ is the input image and $y_{t,i}$ is the ground-truth label (which is utilized only during the validation and test phases). For simplicity, we omit $t$ and only use $x_i, y_i$ in the following subsections. For each input sample $x_i$, we first randomly augment it into two views $x_i^1, x_i^2$. We then consider $f_\theta$ with parameters $\theta$ as the backbone (encoder) and employ it to obtain $d$-dimensional feature representations $\{f_\theta(x_i^1), f_\theta(x_i^2)\} \in \mathbb{R}^d$. Formally, based on a widely-used contrastive learning framework SimSiam (Chen & He, 2021), the main training loss for UCL can be defined as:

$$L_{SimSiam}(f_\theta(x_i^1), f_\theta(x_i^2)) = \frac{1}{2}D(\text{MLP}(f_\theta(x_i^1)), f_\theta(x_i^2)) + \frac{1}{2}D(\text{MLP}(f_\theta(x_i^2)), f_\theta(x_i^1)), \quad (1)$$

where $D(p, z) = -\cos(p, \text{stop\_gradient}(z)) = -\frac{p}{\|p\|_2} \cdot \frac{z}{\|z\|_2}$, and $\text{MLP}(\cdot)$ denotes a multi-layer perception. After training, the model is then evaluated by a K-nearest neighbor (KNN) classifier (Wu et al., 2018) following the setup in (Chen et al., 2020; Chen & He, 2021; Madaan et al., 2021).

Directly applying the SimSiam-based learning loss mentioned above is difficult to obtain a well-performed model due to the catastrophic forgetting problem that the model performance on previous tasks drops significantly during sequential training. To tackle this forgetting problem, rehearsal-based methods with a memory buffer to store limited previous data are most popular solutions in the

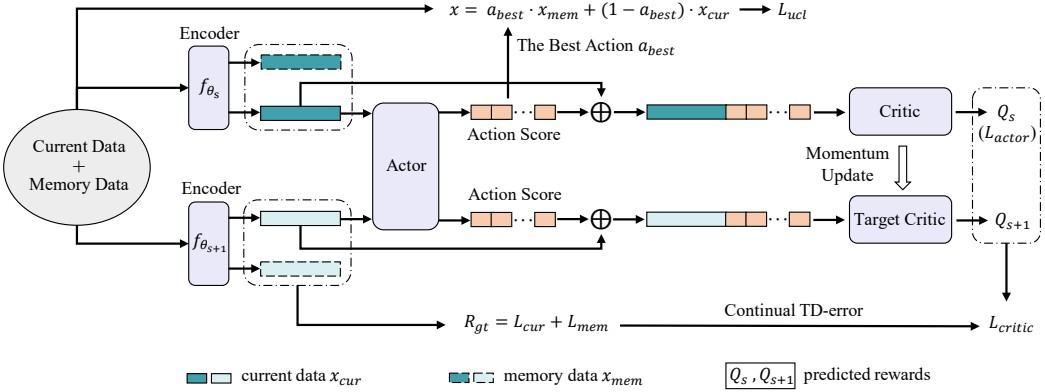

Figure 2: Schematic illustration of our AUDR. At each training step $s$, the Actor selects the best action $a_{best}$ and the Critic predicts the reward of taking this action. We train the Actor with the predicted reward $Q_s$. A reward function $R_{gt}$ with current loss $L_{cur}$ and memory loss $L_{mem}$ of the next-step encoder $f_{\theta_{s+1}}$ is proposed to train the Critic (along with the Target Critic).

past few years. For instance, LUMP (Madaan et al., 2021) proposes a life-long mixup strategy to integrate the memory data with current data as follows:

$$\hat{x}_i^1 = \lambda \cdot x_{m,i}^1 + (1 - \lambda) \cdot x_i^1, \quad \hat{x}_i^2 = \lambda \cdot x_{m,i}^2 + (1 - \lambda) \cdot x_i^2, \tag{2}$$

$$L_{mixup} = \frac{1}{B} \sum_{i=1}^{B} L_{SimSiam}(f_\theta(\hat{x}_i^1), f_\theta(\hat{x}_i^1)), \tag{3}$$

where $B$ is the mini-batch size, $x_{m,i}^1, x_{m,i}^2$ denote two augmentations of the memory data $x_{m,i}$, and $\lambda \sim \text{Beta}(\alpha, \alpha), \alpha \in (0, \infty)$. This strategy is fixed across all tasks and thus it easily leads to sub-optimal performance. In addition, randomly changing the update direction at each training step is not intuitive. In our opinion, whether to use more memory data (i.e., larger $\lambda$) or more current data (i.e., smaller $\lambda$) should be measured by an appropriate pattern. Therefore, in the following, we propose to adaptively rectify the update direction by an Actor-Critic architecture.

### 3.2 ACTION PREDICTION WITH THE ACTOR

We present the schematic illustration of our AUDR in Figure 2. In contrast to randomly sampling the mixup ratio from a fixed Beta distribution, we utilize a policy network (i.e., the Actor) to predict the score distribution of all actions. Concretely, we first define an action space $\mathcal{A} = \{a_j | j = 1, 2, \cdots, N_{act}\}$ where each action $a_j$ denotes different values of $\lambda$ and $N_{act}$ denotes the number of actions. Further, we adopt an Actor $\pi_\phi : \mathbb{R}^d \to \mathbb{R}^{N_{act}}$ to predict and select a best action $a_{best}^i$ based on the representations of the current data $x_i$:

$$a_{best}^i = a_{\arg\max_j [\text{softmax}(\pi_\phi(f_\theta(x_i)))]_j}, \tag{4}$$

where $[\pi_\phi(x_i)]_j$ denotes the $j$-th element of $\pi_\phi(x_i)$. For a mini-batch with $B$ data samples, we first predict the best action of each sample by Eq. (4) and then vote for the action that appears the most frequently as $a_{best}$. We take the selected best action $a_{best}$ as the learning strategy of the current training step $s$ to compute the UCL loss:

$$\tilde{x}_i^1 = a_{best} \cdot x_{m,i}^1 + (1 - a_{best}) \cdot x_i^1, \quad \tilde{x}_i^2 = a_{best} \cdot x_{m,i}^2 + (1 - a_{best}) \cdot x_i^2, \tag{5}$$

$$L_{ucl} = \frac{1}{B} \sum_{i=1}^{B} L_{SimSiam}(f_{\theta_s}(\tilde{x}_i^1), f_{\theta_s}(\tilde{x}_i^2)), \tag{6}$$

where $\theta_s$ denotes the parameters of the encoder $f$ at the current training step $s$. Note that the $a_{best}$ is detached (i.e., without gradient) in $L_{ucl}$. Therefore, this UCL loss is only used for training the encoder $f_\theta$. In this paper, we follow recent Actor-Critic learning works (Laskin et al., 2020; Yarats et al., 2020; Mu et al., 2022) to train the Actor for better predicting the actions, whose learning objective is to maximize the reward predicted by a value network (i.e., the Critic) $Q_\varphi : \mathbb{R}^{d+N_{act}} \to$

$\mathbb{R}^1$ . Formally, for each training step $s$, we define the actor loss as:

$$L_{actor} = -\frac{1}{B}\sum_{i=1}^{B} Q_{s,i} = -\frac{1}{B}\sum_{i=1}^{B} Q_{\varphi}(\text{concat}(f_{\theta_s}(x_i), \pi_{\phi}(f_{\theta_s}(x_i)))), \quad (7)$$

where $\text{concat}(\cdot, \cdot)$ is to concatenate the input vectors as shown in Figure 2 and $Q_{s,i}$ denotes the predicted reward at the current training step $s$. Training the Actor with $L_{actor}$ requires the Critic's prediction to be precise and reliable. However, there is no existing environment which provides the ground-truth rewards of input representations. Therefore, we are supposed to design a dedicated environment (or reward function) for our AUDR.

### 3.3 REWARD FUNCTION FOR THE CRITIC

The widely-used concept "environment" in recent Actor-Critic learning works (Laskin et al., 2020; Yarats et al., 2020; Mu et al., 2022) refers to a pre-defined interactive system, which can provide the ground-truth reward to any input state for the agent. Without the environment, the basis for measuring the values of actions is missing. Therefore, in our AUDR, we desvise a reward function $R_{gt}$ to compensate for the lack of the environment in UCL. $R_{gt}$ measures the model performance on both current data (i.e., $L_{cur}$) and memory data (i.e., $L_{mem}$) after taking the action $a_{best}$:

$$L_{cur} = L_{SimSiam}(f_{\theta_{s+1}}(x_i^1), f_{\theta_{s+1}}(x_i^2)), \quad (8)$$

$$L_{mem} = L_{SimSiam}(f_{\theta_{s+1}}(x_{m,i}^1), f_{\theta_{s+1}}(x_{m,i}^2)), \quad (9)$$

$$R_{gt}^{s,i} = -(L_{cur} + L_{mem}), \quad (10)$$

where $f_{\theta_{s+1}}$ denotes the updated model. Note that the training data is still from the same batch. In other words, $R_{gt}^{s,i}$ evaluates how well the model can perform on the same data after updating. Further, we maintain a target Critic $\mathcal{Q}_{\varphi_t}$ to help train the Critic $\mathcal{Q}_{\varphi}$. It is first proposed by TD3 (Fujimoto et al., 2018) to stablize the training process and has been adopted in many recent works (Laskin et al., 2020; Yarats et al., 2020; Mu et al., 2022). Concretely, $\mathcal{Q}_{\varphi_t}$ is initialized by $\mathcal{Q}_{\varphi}$ and then updated by momentum update strategy (i.e., slowly updated by small part of the parameters of $\mathcal{Q}_{\varphi}$), which is also known as exponential moving average (EMA) (Holt, 2004):

$$\varphi_t = m \cdot \varphi + (1-m) \cdot \varphi_t, \quad (11)$$

where $m$ is the momentum coefficient. Similar to the Critic, the target Critic can predict the reward of input features and actions. Differently, the Critic predicts the reward $Q_{s,i}$ for encoder $f_{\theta_s}$ while the target Critic predicts the reward $Q_{s+1,i}$ for encoder $f_{\theta_{s+1}}$:

$$Q_{s+1,i} = \mathcal{Q}_{\varphi_t}(\text{concat}(f_{\theta_{s+1}}(x_i), \pi_{\phi}(f_{\theta_{s+1}}(x_i)))). \quad (12)$$

As directly predicting the whole future reward is difficult, we could measure the difference between two training steps by temporal difference learning (i.e., TD-error) (Watkins, 1989), which is an update rule based on Bellman equation (Bellman, 1966). Since we have re-designed the ground-truth reward for UCL, we thus call the learning objective of the Critic as "continual TD-error". Formally, we define the continual TD-error as follows:

$$L_{critic} = \frac{1}{B}\sum_{i=1}^{B}(Q_{s,i} - (R_{gt}^{s,i} + \gamma \cdot Q_{s+1,i}))^2, \quad (13)$$

where $\gamma$ denotes the discounted factor. During training, the predicted reward will be closer to the ground-truth reward and thus the selected action becomes more reliable.

### 3.4 RESTRICTIONS ON SAMPLING PROBABILITY

Although we can predict the best action of each training step by our AUDR framework introduced above, it still can not guarantee that the predictions are completely accurate, nor that taking other actions will definitely lead to worse performance. In fact, the predicted optimal model updating direction is possibly not global optimal. Therefore, completely relying on the predictions and ignoring other choices will hinder the model's exploration of a better update direction. To address this problem, we propose to sample each action according to the predicted action score (each score has a

range through clamping) by multinomial sampling. Formally, for each sample $x_i$ and each training step $s$, we adjust Eq. (4) and predict the best action as follows:

$$\hat{\pi}_\phi(x_i) = \text{clamp}(\text{softmax}(\pi_\phi(f_{\theta_s}(x_i))), p_{min}, p_{max}), \tag{14}$$

$$a_{best} = a_{\text{MS}(\text{softmax}(\hat{\pi}_\phi(x_i)), 1)}, \tag{15}$$

where $\text{clamp}(\cdot, p_{min}, p_{max})$ denotes clamping each element of the input vector into the interval $[p_{min}, p_{max}]$, and $\text{MS}(\cdot, 1)$ denotes a multinomial sampling strategy that samples one action according to the input score $\hat{\pi}_\phi(x_i)$ and returns the subscript of the selected action. To be more specific, each element of $\text{softmax}(\pi_\phi(f_{\theta_s}(x_i)))$ represents an action's score which will be set to $p_{min}$ if it is smaller than $p_{min}$ and will be set to $p_{max}$ if it is larger than $p_{max}$. Overall, we can obtain the lower-bound sampling probability $\mathcal{P}_{low}$ for each action as follows:

$$\mathcal{P}_{low} = \frac{\exp(p_{min})}{\exp(p_{min}) + (N_{act} - 1) \cdot \exp(p_{max})}, \tag{16}$$

which means that each action has at least the probability of $\mathcal{P}_{low}$ to be sampled. It is calculated by assuming the most extreme case, where only one action's weight (i.e., one element of $\hat{\pi}_\phi(x_i)$) is $p_{min}$ and the others are $p_{max}$ after clamping.

### 3.5 FULL ALGORITHM

There are three main modules in our AUDR framework: the encoder, the Actor, and the Critic. These modules are updated separately at the training phase and promote each other. Concretely, the encoder is learned by minimizing the SimSiam-based UCL loss with an adaptively adjusted mixup strategy. The Actor aims to maximize the prediction score of the Critic. The Critic improves its prediction preciseness through a continual TD-error. For easier and clearer understanding, we summarize the full training process of our AUDR as follows: (1) The Actor selects the best action from the action space by Eq. (14) and Eq. (15); (2) The encoder is updated with the selected action by Eq. (6); (3) The Actor is updated with the predicted reward of the Critic by Eq. (7); (4) The Critic is updated with the continual TD-error by Eq. (13). Note that the Actor-Critic architecture is removed at the test phase and only the encoder is evaluated with a KNN classifier as in other UCL methods. We present the pseudocode of the full algorithm of our AUDR in Appendix A.

Furthermore, it is worth noting that defining the update direction (or the action space) in the mixup style as we did above is not the only way to construct our AUDR. In fact, this adaptive paradigm could be integrated into many other UCL works. For instance, DER (Buzzega et al., 2020) is a representative work of regularization-based methods, which maintains a penalty function to align the representations of old and new models. Let the weight of the penalty function in the final loss be the action, we could define a new action space for our AUDR. We conduct experiments and provide more details in Sec. 4.4 to show that our AUDR+DER obtains great performance improvements over DER, which demonstrates that our AUDR is indeed extensible and versatile.

## 4 EXPERIMENTS

### 4.1 EXPERIMENTAL SETUP

**Datasets.** We train and evaluate our model on three benchmarks for in-dataset UCL: (1) Split CIFAR-10 (Krizhevsky, 2009) contains 5 tasks, with 2 classes (randomly sampled from 10 classes) per task. (2) Split CIFAR-100 (Krizhevsky, 2009) consists of 20 tasks, with 5 classes (randomly sampled from 100 classes) per task. (3) Split Tiny-ImageNet is a subset of ImageNet (Deng et al., 2009) which has 20 tasks with 5 classes per task. We also evaluate our models (pre-trained on Split CIFAR-10 or Split CIFAR-100) on three benchmarks for cross-dataset UCL: MNIST (LeCun, 1998), Fashion-MNIST (FMNIST) (Xiao et al., 2017), and SVHN (Netzer et al., 2011).

**Evaluation Protocol.** We follow recent works (Mirzadeh et al., 2020; Madaan et al., 2021) to establish the evaluation protocol of UCL. Formally, let $\mathcal{A}_{t,i}$ denote the test accuracy of the model on task $i$ after trained on task $t$, where the total number of tasks is $N$. We define two metrics: (1) *Average Accuracy* denotes the average classification accuracy of the model on all learned tasks after sequential training: Accuracy = $\frac{1}{N}\sum_{i=1}^{N}\mathcal{A}_{N,i}$.

Table 1: Comparison results with state-of-the-art methods under the in-dataset UCL setting on three benchmarks: Split CIFAR-10, Split CIFAR-100 and Split Tiny-ImageNet. "Accuracy" denotes average accuracy and "Forgetting" denotes average forgetting. "Multi-Task" is the upper-bound method which is based on multi-task learning. Standard deviation results are shown in brackets.

| Method | Split CIFAR-10 | | Split CIFAR-100 | | Split Tiny-ImageNet | |
|---|---|---|---|---|---|---|
| | Accuracy (↑) | Forgetting (↓) | Accuracy (↑) | Forgetting (↓) | Accuracy (↑) | Forgetting (↓) |
| Multi-Task | 95.76 (±0.08) | – | 86.31 (±0.38) | – | 82.89 (±0.49) | – |
| Finetune | 90.11 (±0.12) | 5.42 (±0.08) | 75.42 (±0.78) | 10.19 (±0.37) | 71.07 (±0.20) | 9.48 (±0.56) |
| PNN (Rusu et al., 2016) | 90.93 (±0.22) | – | 66.58 (±1.00) | – | 62.15 (±1.35) | – |
| SI (Zenke et al., 2017) | 92.75 (±0.06) | 1.81 (±0.21) | 80.08 (±1.30) | 5.54 (±1.30) | 72.34 (±0.42) | 8.26 (±0.64) |
| DER (Buzzega et al., 2020) | 91.22 (±0.30) | 4.63 (±0.26) | 77.27 (±0.30) | 9.31 (±0.09) | 71.90 (±1.44) | 8.36 (±2.06) |
| LUMP (Madaan et al., 2021) | 91.00 (±0.40) | 2.92 (±0.53) | 82.30 (±1.35) | 4.71 (±1.52) | 76.66 (±2.39) | 3.54 (±1.04) |
| Ours | **93.29** (±0.21) | **1.72** (±0.12) | **84.04** (±0.11) | **3.35** (±0.16) | **77.67** (±0.12) | **3.48** (±0.11) |

Table 2: Comparison results with the state-of-the-art methods under the cross-dataset UCL setting. All models are pre-trained on Split CIFAR-10 (or Split CIFAR-100), and then directly evaluated on MNIST, FMNIST, SVHN, and Split CIFAR-100 (or Split CIFAR-10).

| Method | Split CIFAR-10 | | | | Split CIFAR-100 | | | |
|---|---|---|---|---|---|---|---|---|
| | MNIST | FMNIST | SVHN | CIFAR-100 | MNIST | FMNIST | SVHN | CIFAR-10 |
| Multi-Task | 90.69 (±0.13) | 80.65 (±0.42) | 47.67 (±0.45) | 39.55 (±0.18) | 90.35 (±0.24) | 81.11 (±1.86) | 52.20 (±0.61) | 70.19 (±0.15) |
| Finetune | 89.23 (±0.99) | 80.05 (±0.34) | 49.66 (±0.81) | 34.52 (±0.12) | 85.99 (±0.86) | 76.90 (±0.11) | 50.09 (±1.41) | 57.15 (±0.96) |
| SI (Zenke et al., 2017) | 93.72 (±0.58) | 82.50 (±0.51) | **57.88** (±0.16) | 36.21 (±0.69) | 91.50 (±1.26) | 80.57 (±0.93) | 54.07 (±2.73) | 60.55 (±2.54) |
| DER (Buzzega et al., 2020) | 88.35 (±0.82) | 79.33 (±0.62) | 48.83 (±0.55) | 30.68 (±0.36) | 87.96 (±2.04) | 76.21 (±0.63) | 47.70 (±0.94) | 56.26 (±0.16) |
| LUMP (Madaan et al., 2021) | 91.03 (±0.22) | 80.78 (±0.88) | 45.18 (±1.57) | 31.17 (±1.83) | 91.76 (±1.17) | 81.61 (±0.45) | 50.13 (±0.71) | 63.00 (±0.53) |
| Ours | **93.98** (±0.33) | **83.78** (±0.13) | 55.95 (±1.76) | **39.77** (±0.53) | **94.34** (±0.34) | **83.09** (±0.43) | **55.29** (±0.56) | **69.33** (±0.62) |

(2) *Average Forgetting* is the average performance decrease of the model on each task between its maximum accuracy and the final accuracy: Forgetting = $\frac{1}{N-1} \sum_{i=1}^{N-1} \max_{t \in \{1, \cdots, N\}} (\mathcal{A}_{t,i} - \mathcal{A}_{N,i})$.

**Implementation Details.** All baseline methods and our AUDR are implemented based on SimSiam (Chen & He, 2021) with ResNet-18 (He et al., 2016) as the backbone encoder. The Actor and the Critic are both MLP-based structures. Concretely, the Actor has 4 linear layers and the Critic has a similar architecture with one more 3-layer MLP head which is adopted for clipped double Q-learning (Van Hasselt et al., 2016; Fujimoto et al., 2018). We provide more details for the Actor-Critic architecture in Appendix B. To obtain our main results and make fair comparison to recent competitors, we follow LUMP (Madaan et al., 2021) to average the evaluation results over three independent runs with different random seeds. More details are given as follows: (1) At the training phase, all images are randomly augmented into two views by horizontal-flip, color-jitter, gaussian-blur, and gray-scale. (2) We train our model for 200 epochs per task (the same for all baseline methods). (3) The learning rate is set to 0.03 and the memory buffer size is set to 256 (as in all baseline methods). Our action space has 10 actions which are uniformly sampled from $[0, 1]$ while the minimum action score $p_{min}$ is set to 0.08. The source code will be released soon.

## 4.2 COMPARISON TO STATE-OF-THE-ART METHODS

**In-Dataset UCL.** Table 1 shows the accuracy and forgetting results of our AUDR under the in-dataset UCL setting on three datasets: Split CIFAR-10, Split CIFAR-100, and Split Tiny-ImageNet. We compare our AUDR model with recent representative methods including expansion-based method PNN (Rusu et al., 2016), regularization-based methods SI (Zenke et al., 2017) and DER (Buzzega et al., 2020), and rehearsal-based method LUMP (Madaan et al., 2021). Note that Finetune denotes the lower-bound of UCL, which means directly finetuning the model across all the tasks without any continual learning strategies. The upper-bound of UCL is Multi-Task, which is to simultaneously train the model on all tasks and thus it has no forgetting results. All the results for the baseline methods are directly copied from LUMP (Madaan et al., 2021). It can be clearly seen that our AUDR achieves new state-of-the-art results on all three datasets. Particularly, our AUDR outperforms the second-best method LUMP by 1.68% on accuracy and 0.87% on forgetting (com-

Table 3: Ablation study results for our AUDR. Three groups of experiments are constructed to show the impact of different (a) reward functions, (b) action spaces, and (c) samping strategies.

| (a) Different reward functions. | | | (b) Different number of actions. | | | (c) Different sampling strategies. | | |
|---|---|---|---|---|---|---|---|---|
| | Split CIFAR-10 | | | Split CIFAR-10 | | | Split CIFAR-10 | |
| Method | Accuracy | Forgetting | Method | Accuracy | Forgetting | Method | Accuracy | Forgetting |
| None | 91.00 | 2.92 | $N_{act} = 0$ | 91.00 | 2.92 | random | 91.11 | 3.92 |
| $L_{cur}$ | 91.17 | 2.59 | $N_{act} = 5$ | 92.92 | 2.32 | learnable | 91.30 | 3.14 |
| $L_{mem}$ | 89.83 | **0.55** | $N_{act} = 10$ | **93.29** | **1.72** | w/o MS | 91.94 | 2.97 |
| $L_{cur}+L_{mem}$ | **93.29** | 1.72 | $N_{act} = 20$ | 92.62 | 2.94 | w/ MS | **93.29** | **1.72** |

paring average results on all three datasets). Note that the performance of our AUDR is remarkable since it is quite close to the upper-bound (e.g., 84.04 vs. 86.31 on Split CIFAR-100).

**Cross-Dataset UCL.** Table 2 shows the accuracy and forgetting results of our AUDR under the cross-dataset UCL setting. We first train our model on the training set of Split CIFAR-10 (or Split CIFAR-100) and then directly evaluate it on MNIST, FMNIST, SVHN, and Split CIFAR-100 (or Split CIFAR-10). We can observe that: (1) Our AUDR beats the second-best method SI (Zenke et al., 2017) in 7 out of 8 cases and outperforms it by 2.32% in average, which demonstrates that our AUDR has stronger generalization ability. (2) The results of multi-task learning (i.e., Multi-Task) are not the upper-bound under cross-dataset UCL. This suggests that the model's generalization ability can be better enhanced by UCL (e.g., AUDR and SI) than by multi-task learning.

## 4.3 Ablation Studies

There are three core designs in our AUDR to help transfer the Actor-Critic architecture to the UCL setting: (1) a reward function, (2) an action space, and (3) the multinomial sampling strategy. We thus separately conduct experiments to analyze the contributions of them in Table 3.

**Impact of Different Reward Functions.** Table 3a shows the results of our AUDR with different reward functions. Note that we have defined the original reward function in Eq. (10), which has two components: $L_{cur}$ and $L_{mem}$. We can observe that the results obtained by AUDR with $L_{cur}+L_{mem}$ are better than those obtained by AUDR with none reward function (i.e., 1st row) or with only one of $L_{cur}$ and $L_{mem}$ (i.e., 2nd and 3rd rows). Therefore, the reward function should balance the model performance on both old and new tasks for UCL. The reason why the forgetting results of $L_{mem}$ is smaller than $L_{cur} + L_{mem}$ (0.55 vs. 1.72) lies in the much lower accuracy results of $L_{mem}$, which means that it doesn't learn the old knowledge well and thus it has little knowledge to forget.

**Impact of Different Action Spaces.** We present the results of our AUDR with different action spaces (i.e., different number of actions) in Table 3b. We compare models with $N_{act} \in \{0, 5, 10, 20\}$ and find that: (1) Learning to select actions with our AUDR is better than randomly sampling them from a fixed distribution (i.e., 1st row vs. others). (2) More actions do not always lead to better performance (i.e., 3rd row vs. 4th row). In our opinion, learning with more actions means having a more difficult training process for the Actor-Critic module, especially with limited training iterations (i.e., 200 epochs per task for fair comparison with baseline methods).

**Impact of Different Sampling Strategies.** Table 3c shows the results of our AUDR with different sampling strategies. We compare our multinomial sampling (MS) strategy with three other alternatives: random sampling (1st row), a learnable parameter to directly predict the action's value (2nd row), and without our MS strategy (3rd row). We have the following observations: (1) The Actor-Critic module achieves less satisfactory performance without multinomial sampling (MS) strategy (i.e., 3rd row vs. 4th row), which demonstrates the effectiveness of MS. (2) Models with discrete action space are easier to be trained than those with continuous action space (i.e., 2nd row vs. 3rd row). Note that the 2nd row of Table 3c contains the results of our AUDR with a learnable parameter to represent the action. In other words, it has a continuous action space $[0, 1]$, which can be regarded as a space with infinite actions. Training the parameter in this way, its value will have little change between two training steps (e.g., 0.96 and 0.90). However, the optimal actions between these two steps may have a large margin (e.g., 0.96 and 0.16). This hypothetical situation is possible (and common) due to the random sampling and the distribution shift in UCL. Therefore, a discrete action space is more suitable than a continuous action space for our AUDR.

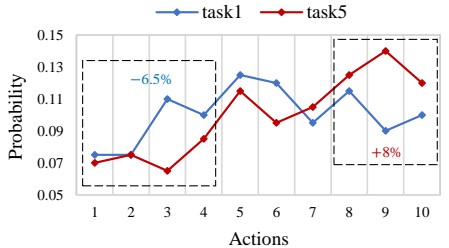 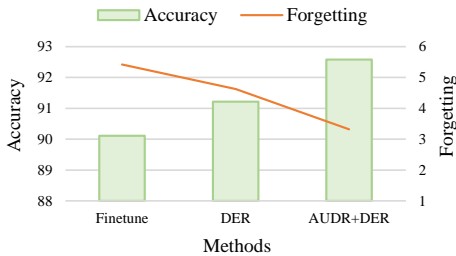

(a) Probability distributions of different actions.    (b) Results for our AUDR+DER on Split CIFAR-10.

Figure 3: Further analysis for our AUDR. (a) The probability distributions of sampled actions for two different tasks (task 1 and task 5) on Split CIFAR-10. (b) Accuracy and forgetting results for our re-designed AUDR with DER (i.e., AUDR+DER) on Split CIFAR-10.

## 4.4 FURTHER ANALYSIS

**Action Distributions of Different Tasks.** To validate the effectiveness of the Actor-Critic module in our AUDR, we draw sampled action distributions of different tasks from Split CIFAR-10. Concretely, we present the results for the first task (task 1) and the last task (task 5) in Figure 3a. Note that the larger the action serial number is, the closer the action value is to 1 (i.e., the coefficient of memory data is higher in Eq. (5)). We can observe that: (1) When training on the last task, our AUDR tends to choose more-higher-value actions than training on the first task (e.g., $8\%$ more actions of $[8, 9, 10]$ and $6.5\%$ less actions of $[1, 2, 3, 4]$). This demonstrates that the model needs to pay more attention to reviewing the old knowledge in the later tasks. (2) Every action on all tasks has chance to be sampled by our AUDR (more than 0.05 in Figure 3a), which is mainly due to our multinomial sampling strategy that holds a lower-bound probability for each action.

**Re-design AUDR with DER.** To demonstrate that our AUDR is extensible in UCL, we change the mixup strategy to another UCL method DER (Buzzega et al., 2020). Note that DER was first proposed for SCL and then re-implemented for UCL in (Madaan et al., 2021). Formally, the UCL loss for DER is defined as follows:

$$L_{DER} = \frac{1}{B} \sum_{i=1}^{B} (L_{SimSiam}(f_\theta(x_i^1), f_\theta(x_i^2)) + \alpha \cdot \|f_\theta(x_{m,i}) - F_{m,i}\|_2), \qquad (17)$$

where $\alpha$ is a fixed coefficient and $F_{m,i}$ denotes the stored features of memory data. The second term of this formula is a penalty function to align the outputs of new and old models but with a fixed ratio $\alpha$. We thus replace $\alpha$ with our predicted action $a_{best}$ to formulate our AUDR+DER:

$$L_{AUDR+DER} = \frac{1}{B} \sum_{i=1}^{B} (L_{SimSiam}(f_\theta(x_i^1), f_\theta(x_i^2)) + a_{best} \cdot \|f_\theta(x_{m,i}) - F_{m,i}\|_2). \qquad (18)$$

Figure 3b shows the results for our AUDR+DER on Split CIFAR-10. More results on Split CIFAR-100 and Split Tiny-ImageNet are also presented in Appendix C. It can be seen that our AUDR+DER outperforms DER by a large margin on both accuracy and forgetting. Such success of AUDR+DER demonstrates that our proposed adaptive paradigm (i.e., AUDR) is highly extensible/versatile and has great potential in the continual learning field.

## 5 CONCLUSION

In this paper, we propose an Actor-Critic framework with adaptive update direction rectification (AUDR) for unsupervised continual learning. We devise three novel designs for our AUDR: (1) A reward function considering the model performance on both current and memory data, which provides reliable ground-truth reward for training the Critic; (2) An action space with discrete actions for the Actor to select; (3) A multinomial sampling strategy to ensure the variance of sampled actions. We conduct extensive experiments on three benchmarks to show that our AUDR achieves new state-of-the-art results for both in-dataset and cross-dataset UCL. Further analysis on action distribution and AUDR+DER demonstrate that our AUDR is indeed effective and extensible.

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

## A FULL ALGORITHM OF AUDR

We provide the pseudocode of the full algorithm for our AUDR in Algorithm 1.

---

**Algorithm 1** Pseudocode of AUDR.

---

**Input:** Encoder $f_\theta$ with parameters $\theta$;
       Actor $\pi_\phi$ with parameters $\phi$;
       Critic $Q_\varphi$ with parameters $\varphi$;
       Target Critic $\mathcal{Q}_{\varphi_t}$ with parameters $\varphi_t$;
       An action space $\mathcal{A} = \{a_j | j = 1, 2, \cdots, N_{act}\}$ of $N_{act}$ actions;
       A dataset $\mathcal{D} = [\mathcal{D}_1, \mathcal{D}_2, \cdots, \mathcal{D}_N]$ of $N$ tasks.
**Output:** The learned $f_\theta^*$
 1: Initialize the Target Critic $\mathcal{Q}_{\varphi_t} = Q_\varphi$;
 2: **for all** task $= 1, 2, \cdots, N$ **do**
 3:    **for all** iteration $s = 1, 2, \cdots, MaxIteration$ **do**
 4:       Sample a mini-batch with $B$ images $\{x_i\}_{i=1}^B$;
 5:       Obtain the best action $a_{best}$ by the Actor with Eqs. (14–15);
 6:       Obtain UCL loss $L_{ucl}$ with Eqs. (5–6);
 7:       Update $f_\theta$ using SGD and obtain $f_{\theta_s}, f_{\theta_{s+1}}$;
 8:       Obtain the predicted reward $R_{s,i}$ with Eq. (7);
 9:       Update the Actor $\pi_\phi$ using SGD;
10:       Obtain ground-truth reward $R_{gt}^{s,i}$ with Eqs. (8–10);
11:       Obtain the target reward $R_{s+1,i}$ with Eq. (12);
12:       Obtain the continual TD-error $L_{critic}$ with Eq. (13);
13:       Update the Critic $Q_\varphi$ using SGD;
14:       Update the target Critic $\mathcal{Q}_{\varphi_t}$ using EMA with Eq. (11);
15:    **end for**
16: **end for**
17: **return** the found best $f_\theta^*$.

---

## B MORE IMPLEMENTATION DETAILS AND DISCUSSIONS

**Details of Actor and Critic.** The structures of our Actor and Critic networks are the same as in DrQv2 (Yarats et al., 2021), where the Actor has a 1-layer trunk network (Linear+Layernorm+Tanh) with one 3-layer MLP head and the Critic has a 1-layer trunk network with two 3-layer MLP heads. The Critic is trained by clipped double Q-learning (Van Hasselt et al., 2016; Fujimoto et al., 2018) to alleviate the over-estimation problem, where two MLP heads represent two Q-functions $Q_{\varphi_1}, Q_{\varphi_2}$ which separately predict the rewards $R_{s,i}^1, R_{s,i}^2$. Then the final predicted reward $R_{s,i}$ of Eq. (7) is obtained by: $R_{s,i} = \min\{R_{s,i}^1, R_{s,i}^2\}$.

**Discussions on Comparing AUDR with Other Possible Methods.** The core idea of our AUDR is to adaptively rectify the update direction during training. In this work, the instantiated "update direction" is based on the mixup ratio (of AUDR) or the penalty loss weight (of AUDR+DER), which is a pre-defined hyper-parameter of the original method (LUMP or DER). In addition to our AUDR, there are three possible approaches to adjusting the update direction during training:
**(1)** Directly using a learnable parameter (through a MLP layer) to represent the update directions. The main drawback of this method lies in the slight change of the action value between two steps, which has already been discussed in Sec. 4.3.
**(2)** Finding the best hyper-parameters by meta-learning (Maclaurin et al., 2015) or reinforcement learning (Cubuk et al., 2019). Different from their objectives of finding the best hyper-parameter combination by training the models several times (each has a whole training process), our AUDR is an online method to adaptively rectify the hyper-parameter (i.e., the update direction) and thus the hyper-parameter is continuously changing during training instead of fixed. Concretely, those hyper-parameter search methods focus on finding an optimal policy for a neural network to solve a specific task (i.e., the best hyper-parameter has fixed value once found). When transferring them to the UCL setting, they would search the action space to find the best (but fixed) ratio. Since each exploration step requires a whole training process, the computation cost of their schema is enormous

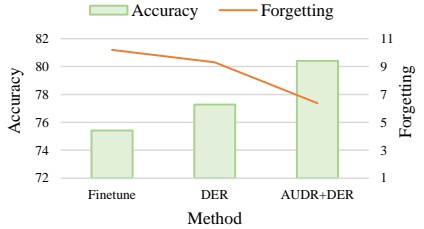

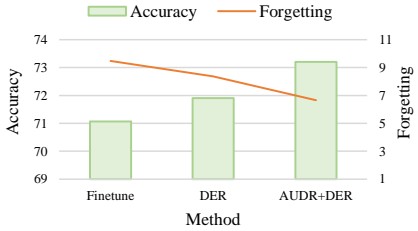

(a) Results for our AUDR+DER on Split CIFAR-100. (b) Results for our AUDR+DER on Split Tiny-ImageNet.

Figure 4: Results for our AUDR+DER on the other two datasets.

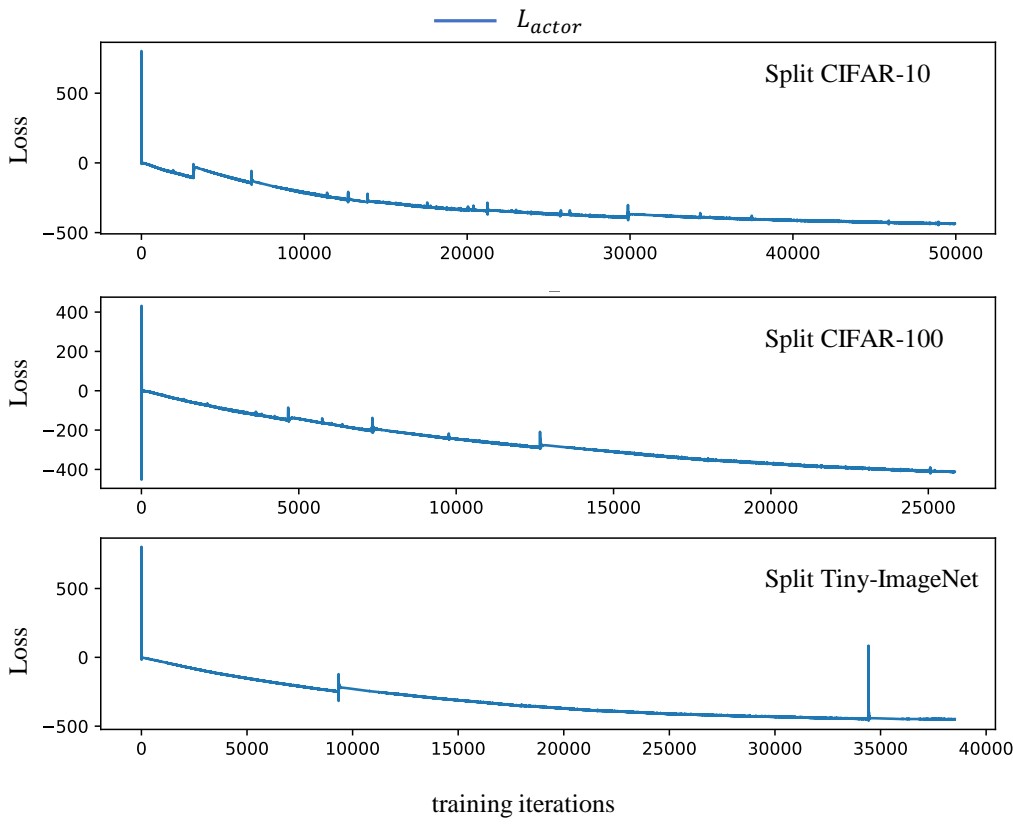

Figure 5: Visualization of Actor Loss during training.

(e.g., 10 trials will lead to 10 times training complexity). The online schema of our AUDR with the Actor-Critic architecture is instead only in need of one whole training process to adjust the mixup ratio at each step, which is significantly more efficient.

**(3)** Learning task-specific prompts (Wang et al., 2022) to be used in the test phase. Differently, the extra Actor-Critic module of our AUDR is removed during testing which is more applicable and resource-saving in real-world application scenarios (e.g., large-scale pre-training).

## C  MORE EXPERIMENTAL RESULTS

We provide a re-designed version of our AUDR in our main paper and present the experimental results on Split CIFAR-10 (see Figure 3b). For a more comprehensive study, we provide the results of our AUDR+DER on Split CIFAR-100 and Split Tiny-ImageNet in Figure 4. We can observe that our AUDR+DER achieves better performance than the competitors in all cases.

Table 4: More results for our AUDR over Split-CIFAR-10. $*$ denotes that our model is trained with 300 epochs per task (otherwise 200 epochs by default).

(a) Different Actor-Critic layers.

| Method | Split CIFAR-10 | |
| | Accuracy | Forgetting |
| --- | --- | --- |
| None | 91.00 | 2.92 |
| 2-layer | 91.47 | 3.55 |
| 4-layer | 93.29 | 1.72 |
| 6-layer | **93.69** | **1.61** |

(b) Longer epochs for 20 actions.

| Method | Split CIFAR-10 | |
| | Accuracy | Forgetting |
| --- | --- | --- |
| $N_{act}$=5 | 92.92 | 2.32 |
| $N_{act}$=10 | 93.29 | **1.72** |
| $N_{act}$=20 | 92.62 | 2.94 |
| $N_{act}$=20$^*$ | **93.60** | 2.61 |

(c) Training with BarlowTwins.

| Method | Split CIFAR-10 | |
| | Accuracy | Forgetting |
| --- | --- | --- |
| Finetune | 87.72 | 4.08 |
| DER | 88.67 | 2.41 |
| LUMP | 90.31 | **1.13** |
| AUDR-BT | **91.53** | 1.98 |

Table 5: Detailed ablation studies. "Max Accuracy" denotes the maximum accuracy on each task over Spilt-CIFAR-10 during training.

| Method | Max Accuracy | | | | | Overall | |
| | Task 1 | Task 2 | Task 3 | Task 4 | Task5 | Accuracy | Forgetting |
| --- | --- | --- | --- | --- | --- | --- | --- |
| $L_{mem}$ | 92.90 | 83.10 | 88.45 | 92.45 | 94.50 | 89.83 | **0.55** |
| $L_{cur} + L_{mem}$ | 97.25 | 91.60 | 92.81 | 95.60 | 96.08 | **93.29** | 1.72 |

## D  VISUALIZATION OF ACTOR LOSS

To verify the training stability of the Actor-Critic architecture applied in our AUDR, we present the plot of the training loss of the Actor in Figure 5. It can be seen that minimizing the actor loss $L_{actor}$ is generally stable during training and finally converges on all datasets. Note that there are relatively large fluctuations during training, which is normal since the distribution of data is constantly changing across sequential tasks.

## E  FURTHER EVALUATION

**Different Actor-Critic Architectures.** We present more results for different Actor-Critic architectures in Table 4a. The MLP-based modules of both Actor and Critic are set to have the same number of layers (0, 2, 4, or 6). We can observe that larger MLP used for Actor/Critic indeed has stronger learning ability and thus leads to better results.

**Training AUDR with More Epochs.** As shown in Table 3b of our main paper, when the number of epochs is limited (e.g., 200 epochs per task), training our AUDR with a larger action space becomes more difficult and thus suffers from performance degradation. To make further verification, we train our model with 20 actions for more (i.e., 300) epochs and present the new results in Table 4b. We find that it leads to better performance, which confirms our assertion.

**Training AUDR with Barlow-Twins.** To show the generalization ability of our AUDR, we change the SimSiam loss with another unsupervised learning loss Barlow-Twins (Zbontar et al., 2021), and then denote our method as AUDR-BT. Experimental results are shown in Table 4c. The results of the competitors are directly copied from LUMP (Madaan et al., 2021). We can observe that our AUDR-BT still achieves the best accuracy on Split CIFAR-10.

**More Detailed Ablation Study.** We provide more detailed results about the ablation study for training our AUDR with only $L_{mem}$ in Table 5. It achieves a much lower overall forgetting rate due to the lower maximum accuracy (i.e., Max Accuracy) on each task. However, it performs significantly worse than training our AUDR with $L_{cur} + L_{mem}$ in terms of overall accuracy.

