# OpenReview forum: "Adaptive Update Direction Rectification for Unsupervised Continual Learning"
_ICLR.cc/2023/Conference — Submitted to ICLR 2023_

### Official Review · Reviewer_GyrZ · 2022-10-22

**Confidence:** 3
**Correctness:** 3
**Technical Novelty And Significance:** 2
**Empirical Novelty And Significance:** 3
**Recommendation:** 6

**Clarity, Quality, Novelty And Reproducibility:**

### Clarity and Quality.
The paper is well-structured, and the main ideas are easy to follow.
### Novelty.
The application of actor-critic to solve URL problems is new. To the best of my knowledge. AUDR or similar framework has not been studied by previous works.
### Reproducibility.
The code has not been uploaded. It is unclear whether the source code will be released. The hyper-parameters for training are unknown. The training complexity requires more study.

**Strength And Weaknesses:**


### Strength

1. The paper is well-structured, and the main ideas are easy to follow.
2. The experiments validate the performance of AUDR from multiple perspectives, and the results are generally promising.
3. The application of actor-critic to solve UCL problems is new. To the best of my knowledge. AUDR or similar framework has not been studied by previous works.

### Weaknesses

1. **Running Complexity.** Training an RL agent often requires multiple runs of explorations and exploitation to update the policy and the value function (e.g., Actor and the Critic networks). Compared to the traditional life-long mixup strategy, AUDR has a larger time and space complexity for building and updating the RL models. The increase in complexity is more significant in a continual learning environment, which often includes multiple tasks (e.g., N tasks in the paper). The planning horizon is very long, compared to a single-task environment.

2. **Continual Reinforcement Learning** I believe the paper has not strictly followed the Continual Reinforcement Learning setting [1], where interventions are not allowed. For example, the game won't end, and movements cannot be reversed. If an action $a$ is performed in a state $s$, we won't know the outcome of other actions, so the vanilla exploration used by traditional RL works will be impossible. The agents must continuously adapt to the environment. If this work does follow this setting, please clarify.

[1] Khimya Khetarpal, Matthew Riemer, Irina Rish, Doina Precup. Towards Continual Reinforcement Learning: A Review and Perspectives.

3. **Some Notations have been misused.**

- $R_{s,i}$ in formula (13) does not indicate reward. In fact, $R_{s,i}$ is the expectation of discounted cumulative rewards by following the current policy (represented by the actor). Please denote it by $Q$ instead of $R$.

- Formula (4) puts a softmax function on top of a policy function. This is abnormal since a policy commonly predicts either the probabilities for discrete actions or the expectation values (assuming the risk-neural policy) for continuous actions. In this work, the action space is discrete, so $\pi$ defines the probabilities for each of the candidate actions. In this case, why a softmax layer is necessary?



**Summary Of The Paper:**

This paper designs an Actor-Critic to learn the optimal mix-up strategy for performing Unsupervised Continual Learning (UCL). The corresponding framework, named AUDR, enables adaptively rectifying the update direction in UCL according to its performance under different mixtures of memory data and incoming data. AUDR includes three major designs including (1) A reward function for measuring the immediate performance after applying a mixture strategy; (2) An action space for representing the mixture strategy; (3) A multinomial sampling strategy based on the softmax function and min-max clamping. The experiments, including several ablation studies under different datasets, empirically demonstrate the performance of the proposed model.

**Summary Of The Review:**

The paper introduces an intriguing framework that enables the application of RL into UCL. The main ideas are well explained. The main claims are correct (to the best of my knowledge). The claims are supported by empirical results. Since the reviewer's background is mainly about RL, the comments are mostly based on the actor-critic design in this work. Although some notations need to be revised, the general quality is satisfying. I am hoping the authors respond to my worries. All in all, I vote for a weak acceptance.

---

> ### Author Response · Authors · 2022-11-14
> **Response to Reviewer GyrZ**
>
> **Q1. Running Complexity. Training an RL agent often requires multiple runs of explorations and exploitation to update the policy and the value function (e.g., Actor and the Critic networks). Compared to the traditional life-long mixup strategy, AUDR has a larger time and space complexity for building and updating the RL models. The increase in complexity is more significant in a continual learning environment, which often includes multiple tasks (e.g., N tasks in the paper). The planning horizon is very long, compared to a single-task environment.**
> \
> **A1.** Good question! As the reviewer stated, our AUDR indeed has a larger time and space complexity compared to LUMP. However, in our opinion, this extra cost is acceptable. Concretely, for each task in Split-CIFAR-100, training our AUDR requires 1.4h (using 5.2GB GPU memory) while training LUMP needs 1.1h (using 4GB GPU memory). Unlike the traditional hyper-parameter search methods that require multiple whole training trials, we only train our model once and rectify the update direction step-by-step with lightweight MLP heads (i.e., MLP-based Actor and Critic). Therefore, the extra cost of our AUDR is limited while achiving much better performance. More importantly, the Actor-Critic module is removed at the testing phase, which means that our AUDR has exactly the same inference efficiency as other methods.
>
> **Q2.Continual Reinforcement Learning. I believe the paper has not strictly followed the Continual Reinforcement Learning setting, where interventions are not allowed. For example, the game won't end, and movements cannot be reversed. If an action $a$ is performed in a state $s$, we won't know the outcome of other actions, so the vanilla exploration used by traditional RL works will be impossible. The agents must continuously adapt to the environment. If this work does follow this setting, please clarify.**
> \
> **A2.** Yes, we do not follow the Continual Reinforcement Learning (RL) setting. In fact, we follow the Unsupervised Continual Learning (UCL) setting, where a model is continuously trained on many classification tasks with unlabeled data. The RL architecture used in our AUDR (i.e., the Actor-Critic) is designed to adaptively rectify the update direction of our model, whose action space and the enviornment remains the same among different tasks (which is different from Continual RL). Overall, our AUDR actually induces RL to UCL (but not following the Continual RL setting).
>
> **Q3. $R_{s,i}$ in formula (13) does not indicate reward. In fact, $R_{s,i}$ is the expectation of discounted cumulative rewards by following the current policy (represented by the actor). Please denote it by $Q$ instead of $R$.**
> \
> **A3.** Thanks. We have exchanged $R$ with $Q$ as the reviewer suggested, and re-polished our notations and figures.
>
> **Q4. Formula (4) puts a softmax function on top of a policy function. This is abnormal since a policy commonly predicts either the probabilities for discrete actions or the expectation values (assuming the risk-neural policy) for continuous actions. In this work, the action space is discrete, so  defines the probabilities for each of the candidate actions. In this case, why a softmax layer is necessary?**
> \
> **A4.** Sorry for the confusion. Indeed, the softmax layer can be removed if we only use Eq. (4) to predict the best action. However, we additionally design a multinomial sampling strategy based on the predicted action score (see Section 3.4). Before sampling, we need to clamp the output score and restrict it with $p_{min}, p_{max}$. Therefore, a softmax layer can help us constrain the score of each action to (0, 1) after which we can easily set the $p_{min}, p_{max}$ to specific values (between 0 and 1).

---

> > ### Comment · Reviewer_GyrZ · 2022-11-26
> > **Reviewer response**
> >
> > Thanks for the clarification. I think the paper is a proper application of RL at UCL. I still think the novelty will be a critical issue in this paper, although my comments are based on the interest of the RL community (maybe the UCL community will be excited about it).

---

### Official Review · Reviewer_6y1n · 2022-10-25

**Confidence:** 2
**Correctness:** 3
**Technical Novelty And Significance:** 3
**Empirical Novelty And Significance:** 3
**Recommendation:** 6

**Clarity, Quality, Novelty And Reproducibility:**

The description of the formulation of the actor critic objective could be more concise. I believe most RL researchers expect reward design to be a challenge specific to each new setting, so instead of explaining why this is challenging in the last paragraph of the introduction and in the first paragraph of section 3.3, I would devote more time to describing what reward function was picked and why. Similarly, instead of describing the target critic update procedure, the authors could reference which actor-critic algorithm they use and refer readers to the corresponding paper.

Miscellaneous points to clarify:
- Is this setting non-episodic? If so, how is the critic inferring the expected cumulative reward (as is typical in RL)? Under the current description, it looks like the critic passes the policy the next step reward to prevent the policy from taking a bad gradient step.
- Should the critic reward be negated? Else the policy is taking actions to maximize the simsiam loss predicted by the critic.
- The DER paper presents DER and DER++ which attains slightly higher performance on some tasks. Which variant is compared against in this work?

**Strength And Weaknesses:**

The greatest strength of the paper is its results: the method attains reasonable performance improvements over baselines, although many are within a standard deviation of other methods. It is also the first work to cast unsupervised continual learning as reinforcement learning to my knowledge. The paper would be improved by demonstrating that the method is applicable across different loss types (e.g., aside from just simsiam). The authors could also compare against P-MNIST, R-MNISt, and MNIST-360 as DER does. I would be comfortable accepting the paper if the method was reliable across different unsupervised learning variants and if the authors address some of the clarification points below.

**Summary Of The Paper:**

The paper devises a strategy for unsupervised continual learning by formulating gradient updates as actions taken by an actor whose loss is determined by a critic approximating the future SimSiam loss value. Specifically, the predictions of the actor determine how much memory and current data are weighted in the encoder SimSiam loss. To increase the diversity of the actor predictions, they clamp the lower bound of the probability of any one action. They evaluate their algorithm on CIFAR-10, CIFAR-100, Tiny-ImageNet, and on a cross dataset evaluation. Their method outperforms the prior art on accuracy and they also show that their method maintains reasonable performance over a couple of different choices of reward functions and action spaces.

**Summary Of The Review:**

Overall I think there are promising results in the paper, but I would appreciate if the authors could address my clarification questions and demonstrate that the method is reliable across different unsupervised learning losses.

---

> ### Author Response · Authors · 2022-11-14
> **Response to Reviewer 6y1n**
>
> **Q1. I would be comfortable accepting the paper if the method was reliable across different unsupervised learning variants and if the authors address some of the clarification points below.**
> \
> **A1.** Good advice. As the reviewer suggested, to show that our method is reliable across different unsupervised learning variants, we conduct more experiments with Barlow-Twins [1] and report the results below:
>
> | Method | Accuracy | Forgetting |
> |:-|:-:|:-:|
> | Finetune | 87.72 | 4.08 |
> | DER | 88.67 | 2.41 |
> | LUMP | 90.31 | **1.13** |
> | AUDR (Ours) | **91.53** | 1.98 |
>
> We have also added these results in Table 4\(c\) of Appendix E. It can be seen that our AUDR achieves the best accuracy compared to other UCL works, i.e., our method is indeed reliable/generalizable (across SimSiam and Barlow-Twins). Note that along with the higher accuracy, we also obtain a slightly worse forgetting rate (compared with LUMP), which is not abnormal and can be explained (see our response to Q8 of Reviewer 1R3g for a similar but more detailed analysis).
>
> [1] Zbontar, J., Jing, L., Misra, I., LeCun, Y., & Deny, S. Barlow twins: Self-supervised learning via redundancy reduction. ICML 2021.
>
> **Q2. The description of the formulation of the actor critic objective could be more concise. I believe most RL researchers expect reward design to be a challenge specific to each new setting, so instead of explaining why this is challenging in the last paragraph of the introduction and in the first paragraph of section 3.3, I would devote more time to describing what reward function was picked and why. Similarly, instead of describing the target critic update procedure, the authors could reference which actor-critic algorithm they use and refer readers to the corresponding paper.**
> \
> **A2.** Thanks for your thoughtful suggestions. Since our AUDR is following the unsupervised continual learning (UCL) setting, researchers who are interested in this work may be experts in continual learning but unfamiliar with RL. Therefore, we pay more attention to describing the RL-related contents and we hope all readers could easily understand our work regardless of with or without RL research experience.
>
> **Q3. Is this setting non-episodic? If so, how is the critic inferring the expected cumulative reward (as is typical in RL)? Under the current description, it looks like the critic passes the policy the next step reward to prevent the policy from taking a bad gradient step.**
> \
> **A3.** Good question. Our setting is actually episodic. Note that typical episode design of other RL works can't be directly transferred to the UCL setting due to the lack of the environment. Therefore, the episodic setting of our AUDR is slightly different. Concretely, we consider each epoch (of each task) as an episode while the Critic infers the expected cumulative reward of it (i.e., we have a done signal at the end of each epoch). The only difference comes from that our model keeps updating its weights across all epochs and thus the output features (i.e., the input states) of the same data from different epochs may be slightly different.
>
> **Q4. Should the critic reward be negated? Else the policy is taking actions to maximize the simsiam loss predicted by the critic.**
> \
> **A4.** Yes, the critic reward is negated. See $L_{actor}$ (i.e., the critic reward) of Eq. (7) in our main paper.
>
> **Q5. The DER paper presents DER and DER++ which attains slightly higher performance on some tasks. Which variant is compared against in this work?**
> \
> **A5.** In this work, we only use DER by following the UCL benchmark proposed in LUMP.

---

### Official Review · Reviewer_kqaK · 2022-10-28

**Confidence:** 3
**Correctness:** 4
**Technical Novelty And Significance:** 3
**Empirical Novelty And Significance:** 3
**Recommendation:** 6

**Clarity, Quality, Novelty And Reproducibility:**

The writing quality of this paper is pretty good. The quality of the experiments is definitely sufficient for work in this area, considering a number of settings and important ablations. The paper contributes to an important emerging area, pushing the boundaries of what can be accomplished without labels. However, the techniques considered are of incremental novelty in light of the previous literature that has considered all of the main ideas before, albeit in slightly different settings. The authors were not able to release their code in time for publication, but promised to make it publicly available shortly.

**Strength And Weaknesses:**

Strengths:
- Leveraging unlabelled information during continual learning could be very impactful with widespread use if done correctly.
- AUDR is shown to outperform the baselines UCL method LUMP in both in-dataset and cross-dataset settings.
- AUDR is shown to be extensible and able to build off other existing continual learning methods like DER.
- The reward function used is shown to add value when trained with a td style critic. I wonder if a simple sum of these two values would more generally be better stated as a weighted combination. However, what the authors have proposed is simple and requires less tuning.
- The authors have highlighted the value of the multinomial sampling strategy in ablations.
- I quite like the idea of using RL approaches to improve supervised continual learning! It reminds me of the discourse on this topic in Khetarpal et al., "Towards Continual Reinforcement Learning: A Review and Perspectives." 2020.

Weaknesses:
- The main innovations of the paper are quite incremental on top of previous work in slightly different domains.
- The paper is mainly driven by high-level intuitions and most design decisions are not really formally justified.
- I am not sure if I read the result in Table 3b as positive. The authors try to highlight this as a contribution, but it seems to me more like highlighting that the approach is brittle to a particular hyperparameter that is not so clear how to set apriori.

**Summary Of The Paper:**

The authors consider the recently proposed setting of unsupervised continual learning (UCL), which I understand as generally studying the extent that unlabelled information can be exploited in a continual learning setting leveraging self-supervised learning. The authors build a new approach, AUDR, extending the recent paper LUMP that uses an actor-critic paradigm to guide learning. The authors note that the three core design features of their algorithm include the proposal of a reward function, action space, and multinomial sampling strategy that lead to improvements in this setting. AUDR is validated with respect to relevant baselines and ablations in both within-dataset and cross-dataset transfer settings.

**Summary Of The Review:**

As highlighted above, I think that there are many strong elements to this paper. I think it explores an important topic and has a pretty solid set of empirical settings and results. I also generally like the idea of using RL to improve continual supervised learning. My main concerns that keep me on the edge about this paper are that the contributions are quite incremental and that the design decisions are not really formally justified within some theoretical framework or conception of the problem. A nagging concern of mine is also that the results may be quite sensitive to hyperparameters given all of the moving parts.

Update After Feedback:

I appreciate the note by the authors about the code and the experiment applying learning for more epochs. This positioning makes more sense to me than in the original case. However, the comment about the main innovations is consistent with my previous understanding. Moreover, I am unsure about the action space design contribution as while it may improve computational efficiency (as the action space size is tied to sample efficiency in RL theory), it is unclear how this balances with the cost of tuning this hyperparameter in general.

---

> ### Author Response · Authors · 2022-11-14
> **Response to Reviewer KqaK**
>
> **Q1. The main innovations of the paper are quite incremental on top of previous work in slightly different domains. The paper is mainly driven by high-level intuitions and most design decisions are not really formally justified.**
> \
> **A1.** Thanks for pointing this out. Our high-level intuitions and designs are based on the observations of specific existing problems (see the third paragraph of the Introduction), and their effectiveness have been supported by extensive empirical results we provide in the Experiments. Moreover, some of our designs have been formally justified such as the restrictions on sampling strategy (Section 3.4) which restrict the lower/upper bound of the probability of sampling each action.
>
> **Q2. I am not sure if I read the result in Table 3b as positive. The authors try to highlight this as a contribution, but it seems to me more like highlighting that the approach is brittle to a particular hyperparameter that is not so clear how to set apriori.**
> \
> **A2.** Good question. We have analyzed the potential reason behind this phenomenon in our main paper that training our AUDR with more actions is harder. Inspired by Reviewer 1R3g (see our response to Q5), we construct an extra experiment by training our model with more epochs per task and provide the resuts in Table 4(b) of Appendix E. The results in Table 4(b) show that training with 300 epochs indeed helps our model (with 20 actions) learn better. This further proves that our conjecture above is correct: training with more actions does not necessarily lead to worse performance, but it is indeed more difficult (at least more epochs are needed).
>
> **Q3. The authors were not able to release their code in time for publication, but promised to make it publicly available shortly.**
> \
> **A3.** We have provided the code in the supplementary material.

---

### Official Review · Reviewer_1R3g · 2022-10-31

**Confidence:** 5
**Clarity, Quality, Novelty And Reproducibility:** I elaborate on all these aspects in t…
**Correctness:** 3
**Technical Novelty And Significance:** 2
**Empirical Novelty And Significance:** 2
**Recommendation:** 6

**Strength And Weaknesses:**

I am well-familiar with the literature and read the full paper in detail. Accordingly, I'll describe the strengths and weaknesses of the paper in the order of originality + quality, clarity, and reproducibility.

## Originality and Quality
### Strengths
* The choice of hyper-parameters plays a vital role in most rehearsal-based methods. The paper proposes an interesting approach for modeling the hyper-parameter search for UCL as an actor-critic framework and would interest the CL community.
* The proposed AUDR framework is flexible and applicable to various prior CL methods, as also demonstrated in the empirical evaluation of the paper.

### Weaknesses
* The paper's objective is primarily a hyper-parameter search for UCL, focusing on the mixup ratio or penalty loss weight. The paper includes a discussion of various hyper-parameter search methods in the appendix. Still, the efficiency of the proposed method over the prior methods needs to be clarified in the current form. The paper claims that it is an "online" method and more efficient than other approaches; however, in this case, it should show a comparison with these methods and highlight the efficiency of the proposed method.
* The proposed framework has also been restricted to two hyper-parameters and can be further strengthened by incorporating other hyper-parameters into AUDR.
* AUDR also requires additional MLP-based architectures for both Actor and Critics. The paper should include ablation and a discussion on the choice of these architectures in the UCL setting.

---

## Clarity
The paper was well-written and easy to follow. I have a few suggestions and clarifying questions:
* While the paper focuses on LUMP, I suggest updating the notations and figures to the proposed AUDR as a general framework applicable to adapt the buffer hyper-parameters of prior CL methods to strengthen the proposed method.
* The paper highlights that more actions only sometimes lead to better performance, possibly due to limited training iterations. I recommend increasing the training epochs for each task to check if that makes the training process more efficient and improves performance.
* The paper should also compare the generated mixup examples using AUDR and LUMP. Additionally, it would be beneficial to include a discussion on the selected actions with high rewards for both LUMP and DER compared to the hyper-parameters used in prior works.
* The paragraph before subsection 3.2 - In addition, it is not … -> The sentence is incomplete.
* Section 4.2, last line of paragraph 1, on on -> on
* The paper mentions that forgetting for $L_{mem}$ is smaller than the combination of losses because it did not learn the old knowledge well. I suggest supporting this statement with an accuracy comparison of prior tasks during training in the appendix.
* The references are inconsistent, NIPS and NeurIPS are randomly used interchangeably, and few articles don't use the conference bibliographies.

---

## Reproducibility
 The code is not provided with the submission. Since the paper is empirical, it is necessary to provide the code to aid the reproducibility of future works.


**Summary Of The Paper:**

The paper focuses on Unsupervised continual learning (UCL) and proposes Adaptive Update Direction Rectification (AUDR), an adaptive learning paradigm for the UCL setting. Mainly, the paper uses an Actor-critical approach, where the actor selects the best action and is updated with the predicted reward by the Critic. It proposes a reward function based on the current task and replay-buffer performance to guide the Critic’s training, which is updated using the continual TD error. During the evaluation, the paper compares the learned encoder with prior methods, showing superior performance on multiple benchmark datasets.

**Summary Of The Review:**

The paper proposes a novel and interesting approach to continual learning; however, the paper can be strengthened further. I am happy to increase my score if the authors address the above concerns. Notably, It lacks a comparison to prior hyper-parameter search approaches and is limited to two choices of hyper-parameters.

---

> ### Author Response · Authors · 2022-11-14
> **Response to Reviewer 1R3g - Part II**
>
> **Q5. The paper highlights that more actions only sometimes lead to better performance, possibly due to limited training iterations. I recommend increasing the training epochs for each task to check if that makes the training process more efficient and improves performance.**
> \
> **A5.** Good advice! We raise the training epochs per task from 200 to 300 and train our AUDR with 20 actions on Split-CIFAR-10. The obtained results are shown below:
>
> | Setting | Accuracy | Forgetting |
> |:-|:-:|:-:|
> | $N_{act}=5$ & Epoch $= 200$ | 92.92 | 2.32 |
> | $N_{act}=10$ & Epoch $= 200$ (Ours) | 93.29 | **1.72** |
> | $N_{act}=20$ & Epoch $= 200$ | 92.62 | 2.94 |
> | $N_{act}=20$ & Epoch $= 300$ | **93.60** | 2.61 |
>
> These results are also added in Table 4(b) of Appendix E. We can observe that increasing the training epochs indeed improves the performance (i.e., Accuracy is increased from $92.62$ to $93.60$). This suggests that including more actions leads to a more difficult training process and thus more epochs are needed. We sincerely thank the reviewer for giving us such insightful advice.
>
> **Q6. The paper should also compare the generated mixup examples using AUDR and LUMP. Additionally, it would be beneficial to include a discussion on the selected actions with high rewards for both LUMP and DER compared to the hyper-parameters used in prior works.**
> \
> **A6.** Since the mixup ratio of our AUDR is continuously changing as we rectify it step-by-step, the generated mixup samples are different from one epoch to another. Therefore, it is hard to compare them with those generated by LUMP (with a fixed ratio). Nevertheless, we have discussed the distribution of the selected actions in our main paper (see Figure 3(a) and Section 4.4). We find that our AUDR tends to choose more higher-value actions (e.g., the mixup ratio) when training on a later task.
>
> **Q7. The paragraph before subsection 3.2 - In addition, it is not … -> The sentence is incomplete; Section 4.2, last line of paragraph 1, on on -> on; The references are inconsistent, NIPS and NeurIPS are randomly used interchangeably, and few articles don't use the conference bibliographies.**
> \
> **A7.** Thanks. We have corrected these typos.
>
> **Q8. The paper mentions that forgetting for Lmem is smaller than the combination of losses because it did not learn the old knowledge well. I suggest supporting this statement with an accuracy comparison of prior tasks during training in the appendix.**
> \
> **A8.** Thanks for the suggestion. We provide the detailed results of training our AUDR with $L_{mem}$ and with $L_{cur}+L_{mem}$ on Split-CIFAR-10 below:
>
> | Method | $Acc_{max}$ on Task1 | $Acc_{max}$ on Task2|$Acc_{max}$ on Task3|$Acc_{max}$ on Task4|$Acc_{max}$ on Task5| Overall Accuracy | Overall Forgetting |
> |:-|:-:|:-:|:-:|:-:|:-:|:-:|:-:|
> |$L_{mem}$ |92.90 |83.10 |88.45 |92.45| 94.50 | 89.83 | **0.55** |
> |$L_{cur}+L_{mem}$ |97.25 |91.60  |92.81 |95.60  | 96.08 | **93.29** | 1.72 |
>
> We have added these results in Table 5 of Appendix E. We can observe that the results do support our statement. For example, the $Acc_{max}$ for $L_{mem}$ of task 2 (i.e., the maximum accuracy on task 2 during training) is much lower than that for $L_{cur}+L_{mem}$ (i.e., 83.10 vs. 90.60). Therefore, even if training with $L_{mem}$ achieves much worse overall accuracy, it's forgetting rate is easily better than training with $L_{cur}+L_{mem}$ (i.e., it makes no sense to evaluate only with forgetting rate).
>
> **Q9. The code is not provided with the submission. Since the paper is empirical, it is necessary to provide the code to aid the reproducibility of future works.**
> \
> **A9.** Thanks. We have provided the code in the supplementary material.

---

> > ### Comment · Reviewer_1R3g · 2022-11-17
> > **Thank you for your response**
> >
> > Thank you for the response; I appreciate the additional experiments conducted during the rebuttal. Especially the ablations conducted for the architectures and epochs are helpful. The authors mention that the paper focuses on the "key" parameters for continual learning; however, other hyper-parameters are equally crucial for continual learning. For instance, Mirzadeh et al. [1] showed the effect of dropout, learning rate with decay, and batch size improving performance over the baselines. Therefore, incorporating these parameters should be helpful for the proposed framework.
> >
> > Overall, the authors have addressed my concerns to a certain extent, and I have revised my scores.
> >
> > [1]  Mirzadeh et al. Understanding the Role of Training Regimes in Continual Learning. NeurIPS 2020

---

> > > ### Author Response · Authors · 2022-11-17
> > > **Further response**
> > >
> > > Thanks for your positive comments and insightful suggestions! We will incorporate more hyper-parameters as the reviewer suggested in our on-going work.

---

> ### Author Response · Authors · 2022-11-14
> **Response to Reviewer 1R3g - Part I**
>
> **Q1. The paper's objective is primarily a hyper-parameter search for UCL, focusing on the mixup ratio or penalty loss weight. The paper includes a discussion of various hyper-parameter search methods in the appendix. Still, the efficiency of the proposed method over the prior methods needs to be clarified in the current form. The paper claims that it is an "online" method and more efficient than other approaches; however, in this case, it should show a comparison with these methods and highlight the efficiency of the proposed method.**
> \
> **A1.** Good question! The hyper-parameter search methods mentioned in Appendix B mainly focus on finding an optimal policy for a neural network to solve a specific task (i.e., the best hyper-parameter has fixed value once found). When transferring them to the UCL setting, they would search the action space to find a best (but fixed) ratio. Since each exploration step requires a whole training process, the computation cost of their schema is enormous (e.g., 10 trials will lead to 10 times training complexity). The online schema of our AUDR with the Actor-Critic architecture is instead only in need of one whole training process to adjust the mixup ratio at each step, which is significantly more efficient. As the reviewer suggested, we have added more discussions in Appendix B to highlight the efficiency of our AUDR.
>
> **Q2. The proposed framework has also been restricted to two hyper-parameters and can be further strengthened by incorporating other hyper-parameters into AUDR.**
> \
> **A2.** Thanks for your advice. The core idea of our AUDR is to adaptively adjust the "key" hyper-parameters of the representative/state-of-the-art continual learning methods, where the "key" hyper-parameters are supposed to play an important role in addressing the catastrophic forgetting problem. For example, LUMP is a rehearsal-based method which utilizes the mixup ratio to control the integration of the old data and the new data; DER is a regularization-based method which devises a penalty loss to align the outputs of old and new models. The most important hyper-parameters of LUMP and DER are the mixup ratio and the coefficient of the penalty loss, respectively. In this paper, we thus design our AUDR to adaptively adjust those key hyper-parameters during training. Indeed, incorporating other hyper-parameters as the reviewer suggested can further strengthen our AUDR, but the action space would accordingly be enlarged and thus it becomes more difficult to obtain a good policy network. We think this is still an open problem, and we will make further exploration in our ongoing work.
>
> **Q3. AUDR also requires additional MLP-based architectures for both Actor and Critics. The paper should include ablation and a discussion on the choice of these architectures in the UCL setting.**
> \
> **A3.** Good advice. The original MLP-based Actor-Critic architecture of our AUDR strictly follows other RL works like DrQ [1] and CURL [2], which consists of a 4-layer Actor and a 4-layer Critic (with a target head). As the reviewer suggested, we add more experimental results for more ablation studies on the choice of these architectures (i.e., 2-layer and 6-layer) and report the obtained results below:
>
> | Method | Accuracy | Forgetting |
> |:-|:-:|:-:|
> | LUMP | 91.00 | 2.92 |
> | 2-layer AUDR | 91.47 | 3.55 |
> | 4-layer AUDR (Ours) | 93.29 | 1.72 |
> | 6-layer AUDR | **93.69** | **1.69** |
>
> It can be seen that MLP with more layers leads to better results due to its higher representation learning ability. We have added these results in Table 4(a) of Appendix E.
>
> [1] Yarats, D., Kostrikov, I., & Fergus, R. Image augmentation is all you need: Regularizing deep reinforcement learning from pixels. ICLR 2020.
> \
> [2] Laskin, M., Srinivas, A., & Abbeel, P. Curl: Contrastive unsupervised representations for reinforcement learning. ICML 2020.
>
> **Q4. While the paper focuses on LUMP, I suggest updating the notations and figures to the proposed AUDR as a general framework applicable to adapt the buffer hyper-parameters of prior CL methods to strengthen the proposed method.**
> \
> **A4.** Thanks for the suggestion. We have shown an example of applying our method to prior CL works (by re-defining the action space and updating Eq. (6) with Eq. (18)), and we could easily follow the similar oprations to transfer our AUDR to prior CL methods. In our opinion, our current notations and figures can better help the reader understand two specific instantiations of our AUDR. However, with minor modifications, our AUDR can be applied to a wide range of prior CL methods, and thus it is indeed a general framework.

---

### Author Response · Authors · 2022-11-14
**General Response: Contributions and New Experiments**

We sincerely thank all reviewers for taking much time and effort to review our paper. We are glad to find that reviewers generally recognized our contributions:

* **Impact.** The paper proposes an interesting approach which would interest the CL community [1R3g]. Leveraging unlabelled information during continual learning could be very impactful with widespread use [kqaK].
* **Model.** The proposed AUDR framework is flexible and applicable to various prior CL methods [1R3g]. The idea of using RL approaches to improve unsupervised continual learning is interesting and novel [kqaK, GyrZ].
* **Experiment.** The experiments validate the performance of AUDR from multiple perspectives, and the results are promising. [1R3g, kqaK, 6y1n, GyrZ].
* **Writing.** The paper is well-written, well-structured, and easy to follow [1R3g, kqaK, GyrZ].

And we also thank all reviewers for their insightful and constructive suggestions, which help a lot in further improving our paper. In addition to the pointwise responses below, we summarize supporting experiments added in the rebuttal according to reviewers’ suggestions.

**New Experiments**

* Training our AUDR with different MLP-based Actor-Critic architectures [1R3g].
* Larger training epochs could help our model learn better with more actions [1R3g, kqaK].
* Training AUDR with other unsupervised losses (i.e., Barlow-Twins) [6y1n].
* More detailed evaluation results on previous tasks [1R3g].

In addition, we have provided the code in the supplementary material. We hope that our pointwise responses below could clarify all reviewers’ confusion and alleviate all of their concerns. We'd like to thank all reviewers’ time again.

---

> ### Comment · Area_Chair_EdV4 · 2022-11-19
> **More clarification on the RL part**
>
> Dear authors, thank you for your answers!
>
> The reviewers acknowledge the importance of this work, the originality of the approach and good experimental results. I think that the paper currently lacks either the intuition as to why the proposed approach works or a clear formulation of the learning task as an RL problem. I think that the points marked by Reviewer 6y1n as 'Miscellaneous points to clarify' are very important for the acceptance of the paper and more clarification is needed.
>
> * Can the learning process considered in the paper be described as a Markov decision process (MDP)?
> What is the state of that MDP and how does the state transition? If the environment is non-stationary, should one follow the setting of Continual Reinforcement Learning as suggested by Reviewer GyrZ?
>
> * If the setting is episodic, you should define the episode in the paper.
>
> * If one epoch of a task forms an episode, we have the knowledge of when one task ends and the next one begins. Are these settings different to the settings of the previous UCL methods such as LUMP? If the settings are different, are the results comparable?
>
> * I agree with Reviewer 6y1n that the definition of the reward (its sign) is confusing. The agent maximizes its reward, while the loss should be minimized. Eq. (10) defines the reward as the loss (small is good).
>
> Minor comments:
>
> * The term "update direction" is used several times in the paper including the title but I do not understand its meaning in the context of the paper.
> * The notation cos(p, q) in Section 3.1 may be confusing as cos() usually denotes the cosine function.

---

> > ### Author Response · Authors · 2022-11-19
> > **Response to Area Chair EdV4**
> >
> > **Q1. Can the learning process considered in the paper be described as a Markov decision process (MDP)? What is the state of that MDP and how does the state transition? If the environment is non-stationary, should one follow the setting of Continual Reinforcement Learning as suggested by Reviewer GyrZ?**
> > \
> > **A1.** Good question. Yes, the learning process in our paper can be described as a Markov Decision Process (MDP). The state of MDP denotes "the parameters of our model". For better understanding, we compare the state transition process of our method and that of other RL works as follows:
> > >**MDP of Other RL works (CURL or DrQv2)**: \
> > Current state (image) $s_t$ -> Encoder (which is learnable and can be updated) -> Obtain feature $F$ -> Actor -> Action $a_t$ -> Obtain the next state $s_{t+1}$ with $s_t$ and $a_t$.
> >
> > >**MDP of Our AUDR**: \
> > Current state (parameters) $s_t$ -> Randomly-sampled data batch (which is changeable and can be updated) -> Obtain feature $F$ -> Actor -> Action $a_t$ -> Obtain the next state $s_{t+1}$ with $s_t$ and $a_t$ (by gradient descent).
> >
> > Therefore, the learning process of our AUDR is similar to that of other RL works, and can be described as a MDP (i.e., $s_1$ is determined by $s_0$ and $a_0$). The only difference lies in that our state is defined as the parameters of our model. In addition, the parameters of our model can be seen as "one sample" of a certain distribution, which has been widely used in meta-learning [a]. As a result, in our setting, each epoch can be regarded as an episode since it satisfies the following conditions among different epochs/episodes: (1) states are from the same distribution (as in meta-learning), and (2) data samples remain the same (i.e., the whole data of each task). Overall, our AUDR is more like a traditional RL setting (with many novel designs to transfer the Actor-Critic architecture to UCL) rather than Continual RL. We will add this clarification in our final version, since there is not much time left to revise our paper in rebuttal phase 1.
> >
> > [a] Finn, C., Xu, K., & Levine, S. Probabilistic model-agnostic meta-learning. NeurIPS 2018.
> >
> > **Q2. If the setting is episodic, you should define the episode in the paper.**
> > \
> > **A2.** Thanks. Our setting does be episodic (see our response to Q1). We will add its definition in our final version.
> >
> > **Q3. If one epoch of a task forms an episode, we have the knowledge of when one task ends and the next one begins. Are these settings different to the settings of the previous UCL methods such as LUMP? If the settings are different, are the results comparable?**
> > \
> > **A3.** Sorry for the confusion. As the common practice in UCL, the definition of "one epoch" means training a model with the whole data of each task one time (both in our AUDR and LUMP). For each task, we train our model 200 epochs as LUMP did. Therefore, our settings are the same as those of the previous UCL methods, and the results are comparable.
> >
> > **Q4. I agree with Reviewer 6y1n that the definition of the reward (its sign) is confusing. The agent maximizes its reward, while the loss should be minimized. Eq. (10) defines the reward as the loss (small is good).**
> > \
> > **A4.** Thanks for reminding us. Our original design of Eq. (10) does be $R_{gt}^{s,i} = -(L_{cur} + L_{mem})$ (please see our source code: line 285 of main.py). We have corrected this typo in our main paper.
> >
> > **Q5. Minor comments: The term "update direction" is used several times in the paper including the title but I do not understand its meaning in the context of the paper.**
> > \
> > **A5.** Note that the action of our AUDR determines the mixup ratio when mixing the current and memory data. Concretely, an action with larger value (e.g., 0.99) means that the memory data accounts for higher percentage when mixed with current data, and thus the "update direction" of our model is more oriented towards improving the model performance on old tasks. Similarly, an action with smaller value (e.g., 0.01) means that the "update direction" of our model is oriented towards improving the model performance on new tasks. Therefore, our model is trained to find a better "update direction" (towards old or new tasks). This has been stated in the fourth paragraph of the introduction section.
> >
> > **Q6. Minor comments: The notation cos(p, q) in Section 3.1 may be confusing as cos() usually denotes the cosine function.**
> > \
> > **A6.** Sorry for the confusion. "cos()" means the cosine similarity function in our work.

---

### Decision · Program_Chairs · 2023-01-20

**Decision:**

Reject

**Justification For Why Not Higher Score:**

Due to the raised concerns, especially the lack of a fundamental framework supporting the proposed algorithm.

**Justification For Why Not Lower Score:**

The experimental results look good.

**Metareview: Summary, Strengths And Weaknesses:**

The paper studies the problem of unsupervised continual learning (UCL) and proposes a new algorithm (named AUDR). The algorithm extends LUMP, an existing algorithm which mixes the views of the current samples with old samples retrieved from memory to prevent forgetting in continual learning. To extend LUMP which samples the mixing coefficient from a fixed distribution, AUDR adjusts the mixing coefficient during training: small values correspond to using memory samples (rehearsal) while large values correspond to learning new tasks. The authors propose a mechanism for automatically adjusting the coefficient using an actor-critic algorithm.

During the discussion, the following points were raised by the reviewers:
- the limited scope of the algorithm (tuning only two hyperparameters and a small number of UCL methods)
- the mixed results obtained by varying some hyperparameters (the number of actions and the number of epochs)
- the lack of a fundamental framework supporting the proposed algorithm.

The authors' responses partially resolved these concerns but a significant amount of uncertainty still remained. Due to the raised concerns, especially the lack of a fundamental framework supporting the proposed algorithm, I recommend rejection.

**Summary Of Ac-Reviewer Meeting:**

During the discussion, the reviewers and AC raised the same concerns that were previously raised on the conference discussion page: the limited scope of the algorithm (tuning only two hyperparameters and a small number of UCL methods), the mixed results obtained by varying some hyperparameters (the number of actions and the number of epochs) and the lack of a fundamental framework supporting the proposed algorithm. The authors' responses partially resolved these concerns but some amount of uncertainty still remained.